# STEERING LLMS' REASONING WITH ACTIVATION STATE MACHINES

## ABSTRACT

Fine-tuning Large Language Models (LLMs) for specialized skills often comes at a steep cost: catastrophic forgetting of their broad general abilities. Activation steering offers a promising alternative, but existing methods are typically stateless, applying a constant intervention that fails to capture the dynamic, history-dependent nature of a reasoning process. We introduce the Activation State Machine (`ASM`), a lightweight dynamic steering mechanism inspired by state-space models from control theory. The `ASM` learns the latent dynamics of an ideal reasoning trajectory from a set of examples and, at inference time, applies real-time corrective interventions to the LLM's hidden states. We demonstrate that `ASM` steering improves zero-shot accuracy across multiple domains, enhancing performance on both mathematical reasoning and physical reasoning. In addition, we show that while supervised fine-tuning incurs a significant performance drop on an unrelated creative writing task, our method preserves over $95\%$ of the base model's fluency measured in perplexity. Our work presents a new paradigm for modular skill injection, enabling the enhancement of specialized capabilities in LLMs without compromising their foundational generality.

## 1 INTRODUCTION

Many applications of Large Language Models (LLMs) require outputs that are not just fluent, but also logically sound and factually consistent, especially in multi-step reasoning tasks (Beaglehole et al., 2025; Ye et al., 2023). This has motivated extensive work on methods to enhance and control the reasoning abilities of LLMs. Two fundamental, often conflicting, requirements emerge in this problem space:

- **Task-Specific Accuracy**: How effectively does the method improve performance on a target reasoning domain, such as mathematics or science?
- **General Capability Preservation**: Does the method enhance the specialized skill without degrading the model's broad, pre-existing abilities (i.e., avoiding catastrophic forgetting)?

Most existing methods succeed on one axis but sacrifice the other, creating a spectrum of solutions with inherent trade-offs. Broadly, they fall into two families: (1) Weight-Modification Methods: Supervised Fine-Tuning (SFT) is the canonical example (Hu et al., 2021). SFT can be highly effective at increasing accuracy on the target task. However, this performance gain comes at a great cost: by permanently altering the model's weights, SFT is well-documented to cause catastrophic forgetting, degrading the model's performance on other, unrelated tasks (Luo et al., 2025). (2) Stateless Steering Methods: Inference-time interventions like activation steering (Beaglehole et al., 2025; Chen et al., 2025) are also non-destructive. These methods typically apply a static "concept vector" to the model's activations at every step. While useful for static attributes like sentiment, this approach is fundamentally misaligned with the nature of reasoning. Reasoning is not a fixed state but a dynamic trajectory, where each step depends causally on the evolving context.

Thus, the current landscape of methods for enhancing reasoning reveals a challenging trade-off between task-specific accuracy and the preservation of general capabilities. What is missing is a method that can balance between improving reasoning accuracy and being non-destructive. 4

We propose the Activation State Machine (`ASM`), a dynamic, stateful steering method that resolves this tension. `ASM` is framed as a lightweight, real-time "navigator" for the LLM's thought process.

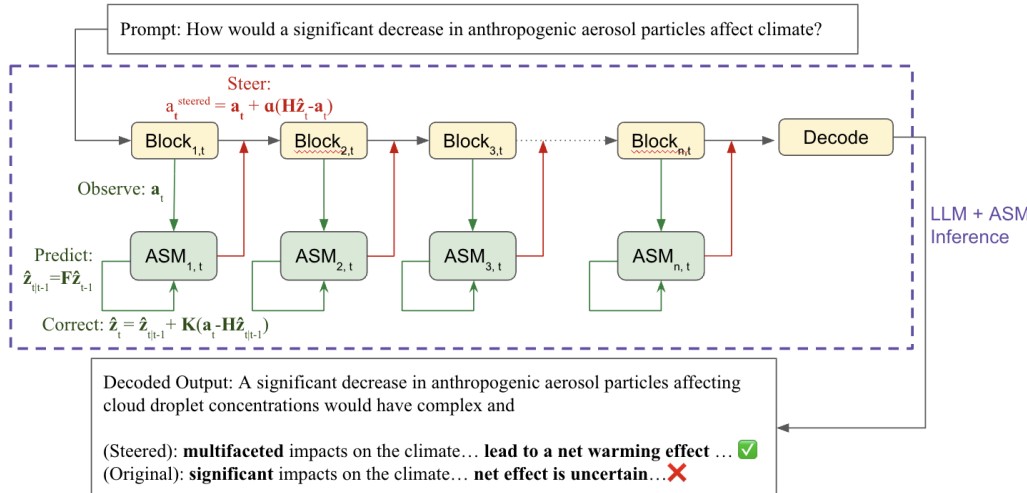

Figure 1: An overview of ASM steering process at inference time. The prompt is fed into the LLM, and for each transformer block being steered, an independent ASM performs a predict-correct cycle. ASM first predicts the ideal state ($\hat{\mathbf{z}}_{t|t-1}$) based on its previous state, then observes the LLM's raw activation ($\mathbf{a}_t$), and finally corrects its internal state ($\hat{\mathbf{z}}_t$) based on the error. This new, smoothed state is used to compute a steering vector that is added to the LLM's activation.

Inspired by control theory, its architecture is a simplified, deterministic form of a Kalman filter (Kalman, 1960), designed to track and guide a dynamic system based on noisy observations. ASMs learn the latent dynamics of an ideal reasoning trajectory from examples. At inference time, ASMs observe the LLM's raw activations and applies a corrective nudge only when needed, keeping the model on a coherent path. This mechanism allows the ASM to be both highly effective and minimally invasive. Our experiments show that ASM achieves a new state-of-the-art in the trade-off between task-specific performance and general capabiilty preservation.

In this work, we make the following contributions:

- **Dynamic reasoning guidance.** We introduce ASM, a lightweight state-machine architecture that adapts steering signals in real time, inspired by deterministic state-space models such as the Kalman filter (Kalman, 1960).

- **Skill injection without forgetting.** Across mathematical and physical reasoning tasks, ASM improves zero-shot accuracy while preserving more than $95\%$ of the model's creative fluency—where fine-tuning severely degrades performance.

- **A new paradigm for modular enhancement.** By enabling reasoning skills to be added without overwriting general abilities, ASM points toward a compositional and non-destructive approach to LLM specialization.

## 2 RELATED WORKS

### 2.1 REASONING IN HIDDEN STATES OF MODERN LLMS

A growing body of research confirms that sophisticated reasoning capabilities are encoded in the hidden states of LLMs. For arithmetic tasks, information from early layers is transmitted to the last token via attention, where late MLP modules then process this information to generate results (Stolfo et al., 2023). For multi-hop problems, intermediate layers form interpretable representations of parallel reasoning paths and potential answers, with feed-forward blocks facilitating the transition to final solutions (Shalev et al., 2024). This observation has led to the concept of "latent thoughts" (Ruan et al., 2025), where the model's internal state represents a more verbose, continuous reasoning process than what is captured in the final text. Methods like Coconut (Hao et al., 2024) and recurrent depth architectures explicitly leverage this by creating recurrent connections in the latent space,

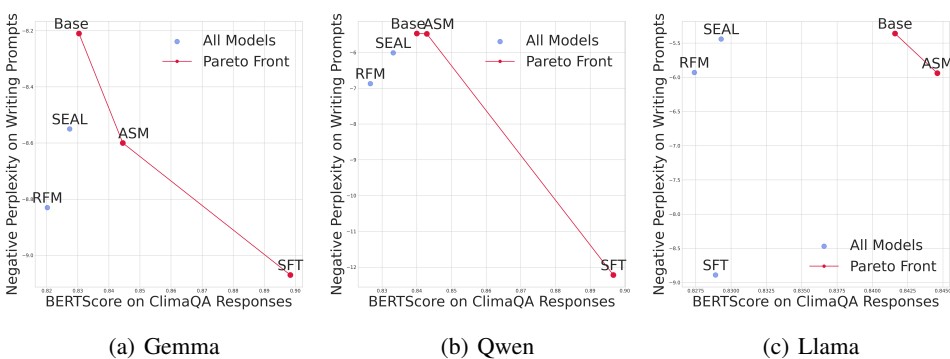

Figure 2: Pareto fronts comparing BERTScore (higher is better) on the ClimaQA dataset and Negative Perplexity (higher is better) on Creative Writing for (a) Gemma-2-9B-it, (b) Qwen2-7B-Instruct, and (c) Llama-3.1-8B-Instruct. We plot negative perplexity vs. BERTScore so that the top-right corner represents the ideal outcome for both metrics. The red line indicates the Pareto front, which is constructed by identifying the set of methods for which no other single method is strictly superior on both axes.

demonstrating that the model's internal state can be treated as a dynamic, evolving system (Yue et al., 2025; Hao et al., 2024; Geiping et al., 2025). This body of work provides the foundational premise for our approach: if reasoning is a dynamic trajectory in the activation space, then a steering method should be able to model and guide that trajectory.

## 2.2 INTERVENTIONS IN THE REASONING PROCESS

**The SFT Paradigm:** The most standard method for teaching a model a new skill is Supervised Fine-Tuning (SFT) (Ouyang et al., 2022). While powerful, SFT directly modifies the model's weights, which often leads to catastrophic forgetting—degrading the model's performance on other, unrelated tasks (Luo et al., 2025). Even modern, parameter-efficient fine-tuning (PEFT) methods like LoRA (Hu et al., 2021) are not immune to this issue (Ren et al., 2024).

**Activation Steering:** Recent approaches in activation steering are designed to offer more fine-grained control. Activation Transport, for example, is a general framework that steers activations guided by optimal transport theory. It accounts for causal relationships across activations by estimating transport maps incrementally for each layer (Rodriguez et al., 2024). Sparse Activation Steering (SAS) leverages sparse autoencoders to steer LLM behavior in sparse spaces for modular control (Bayat et al., 2025). In addition, Conceptors utilize steering matrices for "soft projection" onto a target space, a more nuanced manipulation than simple vector addition, though they can be more computationally expensive and require more data (Postmus & Abreu, 2024).

While these methods push towards greater adaptivity, Recursive Feature Machine (RFM) and Steerable Reasoning Calibration (SEAL) offer specific advancements for reasoning (Beaglehole et al., 2025; Chen et al., 2025). RFM extracts linear representations of general concepts directly from model activations to enable targeted steering and even enhance reasoning capabilities (Beaglehole et al., 2025). SEAL is a training-free method that specifically addresses inefficiencies in Chain-of-Thought (CoT) reasoning by identifying and mitigating redundant reflection and transition thoughts through dynamic interventions in the latent space (Chen et al., 2025). However, while both RFM and SEAL offer powerful inference-time interventions for reasoning, their steering mechanisms typically rely on pre-computed steering vectors. This stateless approach limits their ability to adapt to the evolving context of a complex problem, forcing an unfavorable trade-off between intervention strength and model fluency. In contrast, our stateful `ASM` dynamically computes interventions at each step.

As shown in Figure 2, *the dynamic guidance of `ASM` is the key to achieving a better balance between improving reasoning accuracy without catastrophic forgetting*.

## 3 STATEFUL STEERING WITH ACTIVATION STATE MACHINE

We propose a stateful approach that explicitly models the temporal dynamics of reasoning using a state-space model. We propose a method called **Activation State Machine (ASM) Steering**. This approach models the internal activation dynamics of an LLM as a linear state-space system on a per-layer basis, where we train an independent ASM assigned to each transformer block we wish to steer. This per-layer independence is a crucial design choice motivated by the principle of functional specialization in deep transformers (Kumar et al., 2024). Different layers learn to process information at different levels of abstraction, from syntactic features in early layers to complex semantic and logical relationships in later layers. By training an independent ASM for each layer, we allow each one to become a specialized observer, learning the unique dynamics of information processing at its specific depth. In the following section, we first formally define the ASM's architecture, then detail its use for inference-time steering, and finally describe the training procedure.

### 3.1 ACTIVATION STATE MACHINE: MODEL DEFINITION

The state of our ASM, $\hat{\mathbf{z}}_t \in \mathbb{R}^{d_s}$, is a vector that represents the smoothed, filtered estimate of the LLM's "ideal" reasoning state at time step $t$, where $d_s$ is the dimension of the state space. As illustrated in 1, it first uses its internal model to predict where the ideal reasoning state should go next, then it observes the LLM's actual activation, and finally it uses this observation to correct its own state, ensuring its guidance remains grounded. This forms a corrective feedback loop, which is a simplified, deterministic form of a Kalman filter.

The system is defined by the following components:

- **Observation Vector**: $\mathbf{a}_t \in \mathbb{R}^{d_a}$, which is the raw activation vector from the corresponding LLM layer at time step $t$. $d_a$ is the hidden dimension of the LLM.

- **State Estimate Vector**: $\hat{\mathbf{z}}_t \in \mathbb{R}^{d_s}$, which is the ASM's output.

- **State Transition Matrix**: $\mathbf{F} \in \mathbb{R}^{d_s \times d_s}$, a learned parameter that models the linear dynamics of how the reasoning state evolves.

- **Observation Matrix**: $\mathbf{H} \in \mathbb{R}^{d_a \times d_s}$, a learned parameter that maps the latent state space back to the activation space for comparison.

- **Constant Gain Matrix**: $\mathbf{K} \in \mathbb{R}^{d_s \times d_a}$, a learned parameter that serves as a fixed blending factor, determining how much of the observed error is used to correct the state prediction.

The state update is defined by the following recurrence relation:

$$\hat{\mathbf{z}}_t = \mathbf{F}\hat{\mathbf{z}}_{t-1} + \mathbf{K}(\mathbf{a}_t - \mathbf{H}(\mathbf{F}\hat{\mathbf{z}}_{t-1})). \tag{1}$$

To ensure numerical stability during the recurrent updates, especially over long sequences, we apply spectral normalization (Miyato et al., 2018) to the learned matrices $\mathbf{F}$ and $\mathbf{K}$, which constrains their largest singular value to be at most 1.

### 3.2 TRAINING PROCEDURE

The goal of our training procedure is to learn the parameters $(\mathbf{F}, \mathbf{H}, \mathbf{K})$ of an Activation State Machine for a single, target layer within a specific language model. The input to this process is a dataset, $\mathcal{D}$, which comprises a set of "ideal" reasoning trajectories. Each trajectory, $\{a_t\}_{t=1}^T$, is a sequence of hidden state activations recorded from the target layer of the LLM as it processes a correct "prompt+answer" sequence from a reasoning benchmark.

We frame the training as a form of imitation learning, where the objective is to minimize the one-step prediction error over these recorded trajectories. As illustrated in 1, the ASM observes an activation sequence $\{a_t\}$ and updates its internal estimate $\{\hat{z}_t\}$. The training process, detailed in 1, adjusts ASM's parameters so that the state estimate at one step, $\hat{z}_t$, becomes a good predictor of the next observed activation in the trajectory, $a_{t+1}$. This procedure is repeated independently for each layer we wish to steer, allowing each ASM to learn the activation dynamics present at its specific depth.

---

**Algorithm 1** Activation State Machine Training

---

1: **Input:** Dataset of ideal activation trajectories $\mathcal{D} = \{\{\mathbf{a}_t\}_{t=1}^T\}_i$ where $T$ is the number of tokens.
2: **Input:** Learning rate $\eta$, Number of epochs $N_{epochs}$
3: **for** epoch = 1 to $N_{epochs}$ **do**
4:     **for** each trajectory $\{\mathbf{a}_t^{(i)}\}_{t=1}^T$ in $\mathcal{D}$ **do**
5:         $\hat{\mathbf{z}}_0^{(i)} \leftarrow \text{Initialize}(\mathbf{a}_0^{(i)})$
6:         $\hat{\mathbf{z}}_t^{(i)} \leftarrow \mathbf{F}\hat{\mathbf{z}}_{t-1}^{(i)} + \mathbf{K}(\mathbf{a}_t^{(i)} - \mathbf{H}(\mathbf{F}\hat{\mathbf{z}}_{t-1}^{(i)}))$ for $t = 1, \ldots, T$
7:         $\hat{\mathbf{a}}_{t+1}^{(i)} \leftarrow \mathbf{H}\hat{\mathbf{z}}_t^{(i)}$ for $t = 0, \ldots, T-1$
8:         $\mathcal{L} \leftarrow \frac{1}{T}\sum_{t=0}^{T-1}\|\hat{\mathbf{a}}_{t+1}^{(i)} - \mathbf{a}_{t+1}^{(i)}\|^2$
9:         $(g_{\mathbf{F}}, g_{\mathbf{H}}, g_{\mathbf{K}}) \leftarrow \nabla_{\mathbf{F}, \mathbf{H}, \mathbf{K}}\mathcal{L}$
10:         Update $\mathbf{F}, \mathbf{H}, \mathbf{K}$ using $g_{\mathbf{F}}, g_{\mathbf{H}}, g_{\mathbf{K}}$

---

### 3.3 Inference-Time Steering

At inference time, ASMs are attached to their respective transformer layers using forward hooks. As the LLM generates its response token by token, each ASM observes the LLM's raw activation, updates its internal state, and applies a corrective steering vector $\alpha * (H * \hat{z}_t - a_t)$, where $\alpha$ is a hyperparameter controlling the strength of steering. This vector provides a corrective nudge, gently pushing the LLM's internal state away from its potentially flawed path and back towards the ideal trajectory learned during training. This intervention is applied at each steered layer before the activation is passed to the next component in the transformer block, as detailed in Algorithm 2.

---

**Algorithm 2** Inference-Time Steering with ASM

---

1: **Input:** Pre-trained ASM parameters $\mathbf{F}_l, \mathbf{H}_l, \mathbf{K}_l$ for each steered layer $l$
2: **Input:** LLM, initial prompt, steering strength $\alpha$
3: $\mathbf{a}_{l,0} \leftarrow \text{LLM.get\_activation(prompt)}$     // Get prompt activations for each steered layer.
4: $\hat{\mathbf{z}}_{l,0} \leftarrow \text{Initialize}(\mathbf{a}_{l,0})$     // Initialize the state for each layer from its prompt activation.
5: **for** each token generation step $t = 1, \ldots, N$ **do**
6:     **for** each steered layer $l = 1, \ldots, L$ **do**
7:         $\mathbf{a}_{l,t} \leftarrow \text{LLM.get\_activation}()$
8:         $\hat{\mathbf{z}}_{l,t|t-1} \leftarrow \mathbf{F}_l\hat{\mathbf{z}}_{l,t-1}$
9:         $\hat{\mathbf{z}}_{l,t} \leftarrow \hat{\mathbf{z}}_{l,t|t-1} + \mathbf{K}_l(\mathbf{a}_{l,t} - \mathbf{H}_l\hat{\mathbf{z}}_{l,t|t-1})$
10:         $\mathbf{a}_{l,t}^{\text{steered}} \leftarrow \mathbf{a}_{l,t} + \alpha(\mathbf{H}_l\hat{\mathbf{z}}_{l,t} - \mathbf{a}_{l,t})$
11:         $\text{LLM.set\_activation}(\mathbf{a}_{l,t}^{\text{steered}})$
12:     $\text{token}_{t+1} \leftarrow \text{LLM.generate\_token}()$

---

**Computational Complexity.** The computational overhead of our steering method is minimal, consisting of a series of small, fixed-size matrix operations for each token generated and for each layer being steered. This additional computation is encapsulated within the loop in 2 (lines 4-10). The cost is dominated by four matrix-vector multiplications: one for the state prediction ($F$ matrix, line 6), two for the state correction ($H$ and $K$ matrices, line 7), and one for computing the final steering vector ($H$ matrix, line 8).

Let $d_a$ be the LLM's activation dimension and $d_s$ be the ASM's state dimension. The total complexity of these operations per token, per layer is $O(d_s^2 + 3d_ad_s)$. This cost is constant with respect to the sequence length.

## 4 Experiments

Our experiments are designed to evaluate the effectiveness of the Activation State Machine (ASM) in improving the zero-shot reasoning capabilities of modern Large Language Models, including gemma-2-9b-it (Team et al., 2024), Llama-3.1-8b-Instruct (Grattafiori et al., 2024), and Qwen2-7B-Instruct (qwe, 2024). We test our method on two distinct reasoning domains: mathematical

reasoning and physical reasoning, using GSM8k (Cobbe et al., 2021) and ClimaQA (Manivannan et al., 2025). We train ASMs on the middle to final layers of each language model on each dataset for 30 epochs. Then, we perform a sweep over the steering strength hyperparameter, $\alpha$, to identify the optimal configuration for our evaluation. The best-performing configuration for each model and task is reported in our results. All experiments reported were conducted on NVIDIA A100 GPUs.

We compare our method against a carefully selected set of baselines to evaluate its performance along different axes of intervention. We include Supervised Fine-Tuning (SFT) (Hu et al., 2021) as it is a standard paradigm of teaching a model a new skill. To compare against other inference-time steering methods, we include Recursive Feature Machine (RFM) (Beaglehole et al., 2025) and SEAL (Chen et al., 2025). RFM is a representative example of a stateless steering technique, where a single, static concept vector is used for intervention. SEAL represents the state-of-the-art in training-free reasoning calibration, which also applies a static intervention to guide the model's latent thoughts.

Table 1: Evaluated accuracies on the GSM8k mathematical reasoning benchmark, with methods grouped by intervention type.

| Method | Gemma-2-9B-it | Qwen2-7B-Instruct | Llama-3.1-8B-Instruct |
|---|---|---|---|
| *Prompting Methods* | | | |
| Zero Shot | 0.7544 | 0.8006 | 0.7642 |
| CoT | 0.7619 | 0.8258 | 0.8788 |
| *Weight-Modification* | | | |
| SFT | 0.7589 | 0.7710 | 0.7498 |
| *Inference-Time Steering* | | | |
| RFM | 0.5985 | 0.7273 | **0.8636** |
| SEAL | 0.7273 | **0.8636** | 0.8030 |
| ASM | **0.7703** | 0.8052 | 0.7718 |

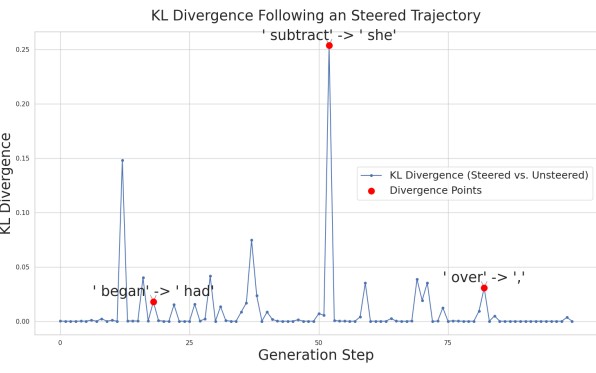

Figure 3: The plot shows the KL Divergence between the steered and unsteered logit distributions at each step of the generation process for a single example. "Divergence Points" indicate moments where ASM intervention had caused the model to choose a different token.

**GSM8k:** Our experimental results on the GSM8k benchmark demonstrate the effectiveness of ASMs. As shown in Table 1, ASMs provide a reliable performance improvement over the Zero Shot baseline across all three models tested. While Chain of Thought (CoT) prompting (Wei et al., 2022) proves to be a particularly strong baseline, especially for the Llama-3.1 model, our ASM shows competitive results. Notably, our method compares favorably to Supervised Fine-Tuning (SFT), which yields surprisingly poor performance on both the Qwen2 and Llama models. It is possible that modern instruction-tuned models have already been exposed to the GSM8k training set, and further fine-tuning on this data can disrupt their well-calibrated reasoning abilities rather than enhance them. *In this context, our ASM's success is particularly significant*; by operating on the latent dynamics of activations, it offers a distinct and effective learning signal, proving to be a reliable paradigm for modular skill injection than re-training on potentially saturated data.

A key observation we made while analyzing what makes a steered sequence correct is that the most impactful interventions are not small, constant nudges, but large, targeted corrections at specific moments in the reasoning process. We show an example below:

**Problem:** Candice put 80 post-it notes in her purse before she headed out to her job at the coffee shop. On her way, she stopped off at the store and purchased a package of Post-it notes; At work, she placed a single Post-it note on each of 220 different cups of coffee. If she had 23 post-it notes remaining overall, how many Post-it notes were in the package that she purchased?

**Unsteered Response:** Here's how to solve the problem:
Start with the total: Candice began with 80 Post-it notes.
Subtract the used notes: She used 220 Post-it notes at work.
Calculate the difference 80 - 220 = -140. This means she was short 140 Post-it notes.
Add the remaining notes: Since she had 23 left, she must have had -140 + 23 = -117 Post-it notes before buying more.
Find the package amount: She needed to buy 117 Post-it notes to make up for the shortage.
Answer: Candice purchased a package of 117 Post-it notes.

**Steered Response:** Here's how to solve the problem:
Start with the total: Candice had 80 Post-it notes + the package she bought.
Subtract the used notes: She used 220 Post-it notes, so she had 80 + package - 220 left.
Set up an equation: We know she had 23 left, so 80 + package - 220 = 23
Solve for the package:
* Combine the constants: -140 + package = 23
* Add 140 to both sides: package = 163
Answer: Candice bought a package of 163 Post-it notes.

A key intervention in this example is the subtle but profound semantic shift from the unsteered model's "Candice began with 80 Post-it notes" to the steered model's "Candice had 80 Post-it notes." The former frames 80 as a fixed total, trapping the model in a flawed subtraction-first reasoning path. The latter creates a more flexible representation of the initial state, allowing the model to correctly incorporate the unknown "package" variable and form a valid algebraic equation. This is confirmed by the KL divergence plot in Figure 3, which shows that for most of the generation, the divergence is near-zero, indicating the ASM is not disrupting the model's natural fluency. However, at a few critical "Divergence Points," ASM can also apply a strong corrective force. This combination of minimal intervention with high-impact corrections at key moments gives our method a decisive edge, allowing it to preserve the model's fluency while ensuring the final answer is correct.

**ClimaQA:** Our results on the ClimaQA physical reasoning task demonstrate the effectiveness of ASMs as a dynamic steering method. As shown in Table 2, ASM consistently outperforms other advanced prompting and steering techniques such as RFM and SEAL across all three base models, establishing its superiority as a lightweight intervention. While Chain-of-Thought (CoT) prompting (Wei et al., 2022) achieves slightly higher scores on n-gram overlap metrics like BLEU (Papineni et al., 2002) and ROUGE-L (Lin, 2004), ASMs consistently yields higher semantic similarity as measured by BERTScore (Zhang et al., 2019), suggesting it produces answers that are more semantically aligned with the ground truth. Furthermore, while SFT achieves a higher performance ceiling on Gemma and Qwen2, our ASM attains a stronger result on Llama-3.1, demonstrating its potential as a robust alternative, particularly in scenarios where fine-tuning may not yield optimal performance.

**Analysis of Catastrophic Forgetting:** A primary motivation for our method is to avoid catastrophic forgetting, a key drawback of Supervised Fine-Tuning (SFT) where a model's general capabilities degrade after being specialized on a new task (Luo et al., 2025). To test this, we compare a SFT fine-tuned models against a base model guided by variaous steering method. Both are evaluated on a creative writing task (Fan et al., 2018), with performance measured by perplexity, a standard metric for linguistic fluency where a lower score is better.

As shown in Table 3, across all three base models, the SFT version exhibits a significant increase in perplexity, indicating a degradation of its core language abilities. In contrast, the perplexity of ASM-steered models remain close to that of the un-steered baseline, especially in Qwen and Llama, where the perplexities are almost identical as the original LLM. This provides strong evidence that

Table 2: Evaluation results on ClimaQA, grouped by intervention type. Best overall performance is in **bold**; best among inference-time methods is underlined.

| Method | Metric | Gemma-2-9B-it | Qwen2-7B-Instruct | Llama-3.1-8B-Instruct |
|---|---|---|---|---|
| *Prompting Methods* | | | | |
| Zero Shot | BLEU | 0.0240 | 0.0245 | 0.0280 |
| | ROUGE-L | 0.1174 | 0.1096 | 0.1378 |
| | BERTScore | 0.8304 | 0.8400 | 0.8416 |
| CoT | BLEU | 0.0379 | 0.0363 | **0.0411** |
| | ROUGE-L | 0.1724 | 0.1436 | **0.1514** |
| | BERTScore | 0.8364 | 0.8437 | 0.8407 |
| *Weight-Modification* | | | | |
| SFT | BLEU | **0.2257** | **0.1975** | 0.0373 |
| | ROUGE-L | **0.3676** | **0.3468** | 0.1419 |
| | BERTScore | **0.8984** | **0.8967** | 0.8389 |
| *Inference-Time Steering* | | | | |
| RFM | BLEU | 0.0250 | 0.0234 | 0.0243 |
| | ROUGE-L | 0.1116 | 0.0939 | 0.0973 |
| | BERTScore | 0.8202 | 0.8266 | 0.8274 |
| SEAL | BLEU | 0.0251 | 0.0238 | 0.0260 |
| | ROUGE-L | 0.1331 | 0.1002 | 0.1055 |
| | BERTScore | 0.8274 | 0.8332 | 0.8293 |
| ASM | BLEU | 0.0295 | 0.0276 | 0.0328 |
| | ROUGE-L | 0.1140 | 0.1404 | 0.1428 |
| | BERTScore | 0.8445 | 0.8429 | **0.8446** |

Table 3: Average Perplexity of story generated using Writing Prompts Dataset

| Dataset | Method | Gemma-2-9B-it | Qwen2-7B-Instruct | Llama-3.1-8B-Instruct |
|---|---|---|---|---|
| LLM | Zero Shot | 8.21 | 5.47 | 5.36 |
| ClimaQA | SFT | 9.07 | 12.22 | 8.89 |
| | SEAL | **8.55** | 6.01 | **5.44** |
| | RFM | 8.83 | 6.87 | 5.93 |
| | ASM | 8.60 | **5.82** | 5.94 |
| GSM8k | SFT | 9.61 | 7.88 | 6.03 |
| | SEAL | 11.93 | 11.58 | 10.07 |
| | RFM | 14.53 | 12.03 | 8.08 |
| | ASM | **8.63** | **5.48** | **5.38** |

our method is non-destructive; by operating on activations at inference time rather than permanently altering the model's weights, ASMs act as a modular skill injector that enhances reasoning without sacrificing the model's foundational generality. In addition, we perform a pareto front on the trade-off between task-specific performance (BERTScore on ClimaQA) and general fluency (Perplexity), further confirming this finding.

**Pareto Front Analysis of Perplexity vs. BERTScore** As shown in Figure 2, the Pareto analysis reveals the high cost of SFT. Across all three base models, the SFT version often achieves the highest BERTScore, demonstrating its effectiveness at specializing in the target task. However,

Table 4: Ablation study on KLD-gated steering with a threshold of $\tau = 0.1$. We report the best accuracy achieved with continuous (non-gated) steering versus the best accuracy achieved with conditional, gated steering across the optimal layers for each model on GSM8k.

| Method | Gemma-2-9B-it | Qwen2-7B-Instruct | Llama-3.1-8B-Instruct |
|---|---|---|---|
| Zero Shot | 0.7544 | 0.8006 | 0.7642 |
| Continuous Steering (Best) | **0.7703** | 0.8052 | **0.7718** |
| KLD-gated Steering (Best) | 0.7665 | **0.8089** | 0.7642 |

SFT's specialization comes at the cost of a drastic increase in perplexity on the creative writing task, indicating significant degradation of its core language abilities. In contrast, our ASM-steered model consistently lies on the Pareto front, achieving a BERTScore far superior to the baseline while maintaining a perplexity score that is only marginally higher. This provides strong evidence that our method is non-destructive while achieving state-of-the-art steering results on ClimaQA.

**Ablation Study: The Impact of Conditional Steering** A key hypothesis is that ASM's effectiveness stems from precise interventions at critical moments. To test this, we conducted an ablation study using KLD-gated inference. This method makes steering conditional based on its immediate potential impact. At each generation step $t$, we first calculate the steering vector that ASMs would apply. We then compute the KL Divergence between the model's original output logit distribution and the distribution that would result if we applied the steering vector. The intervention is only actually applied if this KLD value exceeds a fixed threshold, $\tau$. This provides an gating mechanism for steering, activating only when ASMs propose a correction that significantly alters the model's next-token decision.

For Gemma and Llama, performance remains remarkably stable, dropping only marginally from 0.7703 to 0.7665 and 0.7718 to 0.7642, respectively. This demonstrates that a large portion of the low-KLD "gentle nudges" can be filtered out while still retaining most of the performance benefits, suggesting that the high-impact corrections at key "Divergence Points" are the primary drivers of success. Interestingly, for Qwen, the gated approach not only maintains performance but achieves a slight improvement, rising from 0.8052 to 0.8089. This suggests that for some models, the continuous stream of low-impact nudges may introduce a small amount of noise, and a sparse, high-impact intervention strategy can be even more effective. Together, these findings validate our hypothesis that ASM's power lies in its ability to make targeted corrections at critical moments.

## 5 CONCLUSION AND DISCUSSION

In this work, we introduced the Activation State Machine (ASM), a novel, dynamic steering mechanism designed to address the limitations of stateless interventions and the risks of catastrophic forgetting from fine-tuning. By modeling the latent dynamics of an LLM's reasoning process as a state-space system, our method provides a robust way to guide the model along a more coherent trajectory. Our empirical results validate this approach, showing significant zero-shot accuracy gains on both the GSM8k mathematical and ClimaQA physical reasoning benchmarks. Furthermore, our direct comparison with Supervised Fine-Tuning (SFT) confirmed that our modular, inference-time intervention is non-destructive. The ASM thus presents a promising paradigm for modular skill injection, where specialized capabilities can be applied on-demand without permanently altering the foundational model.

We acknowledge several limitations that motivate future work. The current ASM is a linear model, and the optimal layer and steering strength were determined empirically; future work could explore non-linear state models and develop more systematic methods, such as diagnostic probes, for identifying optimal intervention points. Additionally, the current training via backpropagation through time can be less effective for long sequences, suggesting a need for more scalable architectures. We believe addressing these limitations is a significant step towards building more reliable and controllable Large Language Models.

## REPRODUCIBILITY STATEMENT

We are committed to ensuring the reproducibility of our research. All code and scripts required to replicate the experiments presented in this paper will be made publicly available in a GitHub repository upon publication.

**Data:** All datasets, including GSM8k (Cobbe et al., 2021) and ClimaQA(Manivannan et al., 2025), used in our experiments are publicly available.

**Models:** The base Large Language Models used in our experiments: Gemma-2-9B-it(Team et al., 2024), Qwen2-7B-Instruct(qwe, 2024), and Llama-3.1-8B-Instruct(Grattafiori et al., 2024), are all publicly available on the HuggingFace Hub.

**Code:** The repository will include Python scripts for: (1) collecting the activation traces from the base LLMs; (2) training the Activation State Machine (ASM) models; (3) running inference with ASM steering; and (4) evaluating the results for all tasks. The implementation is primarily based on PyTorch and the Hugging Face 'transformers' and 'datasets' libraries.

## ETHICS STATEMENT

The research presented in this paper adheres to the ICLR Code of Ethics. Our work aims to contribute positively to society by developing methods that make Large Language Models more reliable and accurate at complex reasoning tasks. All datasets used are publicly available benchmarks, and all code and models will be released to ensure scientific transparency and reproducibility.

We also acknowledge the potential for dual-use applications. Any method that grants finer control over an LLM's generative process could, in principle, be adapted for malicious purposes, such as generating more coherent or believable misinformation. Furthermore, as our method learns from example data, any biases present in the training datasets could be learned and propagated by the steering mechanism. We believe that the societal benefit of creating more controllable and demonstrably robust reasoning systems is a crucial step towards mitigating the broader risks of powerful AI, and that transparently sharing this research is the most effective path toward developing shared safeguards.

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

## A    DECLARATION OF LLM USAGE

Large Language Models (LLMs) are the object of study in this work. However, no LLM was used as a component of our core proposed methodology, or for any part of the experimental data analysis. We used Google's Gemini Pro as a writing assistant throughout the research process. Its use included refining prose, generating explanatory text for concepts, drafting document outlines, creating figure captions, and assisting with the generation of boilerplate code for data processing and plotting. All final claims, experimental designs, results, and conclusions were conceived and verified by the human authors, who take full responsibility for the scientific content of this paper.

