# OpenReview forum: "Steering LLMs' Reasoning With Activation State Machines"
_ICLR.cc/2026/Conference — Submitted to ICLR 2026_

### Official Review · Reviewer_r11H · 2025-10-26

**Soundness:** 2
**Presentation:** 2
**Contribution:** 3
**Rating:** 4
**Confidence:** 4

**Summary:**

This paper presents a novel method for steering LLM during inference. The core problem this work addresses is the well-known trade-off between task specialization and general capability. On one hand, Supervised Fine-Tuning (SFT) can improve performance on a target task (like reasoning) but often leads to catastrophic forgetting of the model's general abilities. On the other hand, existing inference-time activation steering methods are often "stateless"—applying a constant, static vector—which is poorly suited for the dynamic, step-by-step nature of a reasoning process.

The authors propose ASM as a "stateful" steering mechanism. Inspired by state-space models and control theory (specifically, a deterministic Kalman filter), the ASM is a lightweight model trained to learn the "latent dynamics of an ideal reasoning trajectory" from a set of correct examples. At inference time, the ASM runs in parallel with the LLM. In each layer, it predicts what the "ideal" activation state should be, observes the LLM's actual activation, and then computes a corrective steering vector based on the error (or "innovation") between the two. This nudge is designed to guide the LLM's internal state back toward a correct reasoning path. The paper's main claim is that this dynamic, stateful approach can improve reasoning accuracy on specialized tasks (mathematical and physical reasoning) while, crucially, preserving the model's foundational fluency on unrelated tasks.

**Strengths:**

- The primary strength of this paper is its core idea. It elegantly connects two major, and previously separate, fields of research: 1) state-space models (SSMs) and linear attention architectures, and 2) model intervention/steering. Reframing inference-time steering as a dynamic control theory problem, rather than a static vector addition, feels like a significant conceptual step forward. It aligns far better with the intuitive understanding of reasoning as a temporal, history-dependent process.

- The choice of the model (Equation 1) is a clean and well-motivated. It is, as the authors note, a simplified, deterministic form of a Kalman filter. This formulation is highly reminiscent of gated mechanisms (as you noted, similar to a Gated DeltaNet), where the learned Kalman gain matrix K effectively acts as a dynamic "gate" on the error term $(a_t - H(F\hat{z}_{t-1}))$. This allows the model to learn how much to intervene based on the observed error, rather than intervening by a constant amount at every step. This is a very strong and lightweight design.

- The strongest experimental finding is the preservation of general capabilities, as shown in Table 3 and Figure 2. While SFT and, surprisingly, other steering methods (RFM, SEAL) cause a massive spike in perplexity (i.e., they "forget" how to write creatively), the ASM-steered model's perplexity remains almost identical to the base model. This is a very practical and important result, as it demonstrates a path toward creating modular, "plug-and-play" skills without permanently damaging the base model.

**Weaknesses:**

- The paper's central claim is to improve reasoning, but the results on this front are deeply unconvincing. In Table 1 (GSM8k) and Table 2 (ClimaQA), the performance of ASM is, at best, trivially better than the zero-shot or CoT baselines. In several cases (e.g., Qwen2 and Llama-3.1 on GSM8k), ASM is outperformed by standard CoT prompting. This lack of a clear accuracy improvement over simple, no-training baselines seriously undermines the practical utility of the method.

-  While the choice of a linear state-space model (Equation 1) is "elegant," the paper provides no justification for why this specific form is appropriate. The space of potential "compression" designs for historical context is vast. Why this linear recurrence? The latent trajectory of "ideal reasoning" is almost certainly a highly complex, non-linear process. The authors don't ablate this design choice or provide analysis on why this simple linear model is sufficient, or if a non-linear model would be better. The analysis on this front is very shallow.

-  For a paper claiming to enhance reasoning, the choice of benchmarks (GSM8k and ClimaQA) is quite weak. These are well-studied, common benchmarks that are likely saturated or at least familiar to the base models. Competing work (like SEAL, which they compare against) is often evaluated on much more difficult reasoning benchmarks like AIME or the MATH dataset. Showing marginal gains on these "easier" datasets does not make a strong case for the method's power.

- Wierd Baseline Performance and Lack of Analysis:
   - **The SFT Results**: the SFT baseline performing worse than the zero-shot baseline in Table 1 is highly suspect. The paper's explanation—that the models have already seen the data—is a possible but convenient excuse. It makes it difficult to fairly evaluate the "accuracy vs. forgetting" trade-off when the SFT baseline isn't even demonstrating high accuracy.

   - **The Perplexity Analysis**: Table 3 is a huge missed opportunity. While it shows ASM is good, it also shows that RFM and SEAL are disastrous for fluency, with perplexity scores doubling or tripling. The paper just presents this data and moves on. A deep analysis of why these static steering methods cause such catastrophic failure on an out-of-distribution task would have been a fantastic contribution. The paper fails to dig into why its dynamic method avoids this, which is a key part of its own story.

- IMO I would split intervention research into learnable intervention and test-time intervetion, I generally believe this works belongs to a learnable option, so I would recommend authors include two related work for learnable intervention:
  - Wu, Zhengxuan, et al. "Reft: Representation finetuning for language models." Advances in Neural Information Processing Systems 37 (2024)
  - Deng, Chunyuan, Ruidi Chang, and Hanjie Chen. "Learning Distribution-wise Control in Representation Space for Language Models." Forty-second International Conference on Machine Learning.

**Questions:**

- Given that the performance gains on GSM8k and ClimaQA are quite close to the CoT baselines, could you elaborate on the specific types of reasoning problems or failure modes where you believe ASM offers a distinct advantage that isn't captured by these aggregate metrics? For instance, does it show more robustness on longer, multi-step problems?

- The choice of a linear state-space model (Equation 1) is elegant, but the 'ideal' reasoning trajectory is likely a complex, non-linear process. Have you experimented with non-linear ASMs, and what are your thoughts on the trade-off between the current model's simplicity and its capacity to correct more complex deviations in reasoning?

- Table 3 provides a fascinating insight: static steering methods (RFM, SEAL) catastrophically degrade fluency, while the dynamic ASM does not. What is your hypothesis for the mechanism behind this? Does the ASM learn to "turn off" intervention by driving the gain matrix K towards zero during fluent generation, only applying strong corrections at critical reasoning forks?

- How do you expect the ASM to perform on more challenging reasoning benchmarks like AIME or MATH, where the 'ideal reasoning trajectories' are far more complex? Would the linear model be sufficient, or would it require a different ASM design to manage the increased complexity?

---

### Official Review · Reviewer_yd1u · 2025-10-31

**Soundness:** 2
**Presentation:** 2
**Contribution:** 2
**Rating:** 4
**Confidence:** 3

**Summary:**

This paper introduces Activation State Machine (ASM), a lightweight method to steer the reasoning behavior of large language models. ASM is trained to intervene model's hidden stages from labelled reasoning trajectories. At test time, ASM corrects the model states to affect the model's behavior. Experiments demonstrate that ASM improves the reasoning performance compared to other baselines, while preserving the base model's ability on an unrelated creative writing task.

**Strengths:**

1. The paper is well structured.
2. The idea is simple and straightforward.

**Weaknesses:**

1. The motivation of formatting ASM as a Kalman filter (Eq. (1)) is not clear. Though in Section 2.2 it's said that "stateless approach limits their ability to adapt to the evolving context of a complex problem", the authors do not explain why the formulation of Kalman filter is a good choice for "state-machine" steering.
2. Experiment setting is ambiguous. Important details of the experiment, such as training data, are omitted by the authors, making it hard to faithfully access the proposed method. Also, it's a bit weird that SFT performance worse than Zero-shot, according to the results in Table 1.
3. Limited evaluation of the proposed method. The main experiments are only done for two datasets: GSM8K and ClimaQA. And GSM8K is relatively simple for the current model. A more comprehensive evaluation on diverse datasets would strengthen the results of the paper.
4. Ablations are missing. The design like the choice of activation layer is not justified.

**Questions:**

1. What are the roles of Matrix F, H, K in Eq. (1), respectively? Would a simpler format (e.g., with only two learnable matrix) perform worse or better?
2. What's the exact computation cost of ASM at inference time, e.g., time or FLOPs?
3. In Table 1, why SFT performs worse than zero-shot?

---

### Official Review · Reviewer_BKmY · 2025-11-01

**Soundness:** 3
**Presentation:** 3
**Contribution:** 3
**Rating:** 4
**Confidence:** 3

**Summary:**

The paper proposes a new inference-time steering method named Activation State Machines that improves LLM reasoning without harming general abilities.  ASM treats reasoning as a trajectory in activation space and applies Kalman filtering to track the latent reasoning state. At training stage, ASM is learned per layer from activation traces of reasoning examples. At inference time, the proposed algorithm will add the minimal steering vector to correct the reasoning trajectory. Experimental results shows promising performance gain, where ASM lies roughly on the pareto frontier comparing BERTScore on the downstream task dataset and Negative Perplexity on another unrelated task while maintains comparable accuracy on the target task against former related works.

**Strengths:**

- The paper is overall well-written and understandable.
- The proposed state machines style methods is highly novel in the LLM steering regime.
- The experimental results are overall solid comparing with previous works.

**Weaknesses:**

- The experiments are limited about steering towards only two reasoning tasks (GSM8k and ClimateQA), which is too narrow.
- The training stage seems to be burdensome,so it is possibly unfair to compare ASM with training-free steering methos like SEAL.
- The pareto front comparison does not show that ASM is significantly better than SEAL and RFM. All three methods are actually quite close to each other since the horinzontal axis is displayed in small granularity.

**Questions:**

Questions are addressed in the Weaknesses section.

---

### Meta-Review · Area_Chair_7ev5 · 2026-01-06

**Summary:**

This paper proposes a new method for steering reasoning in LLMs using a steering mechanism trained on the latent dynamics of reasoning trajectories of training examples.  The reviews were all borderline reject (4, 4, 4) though the reviewers noted a number of positive aspects, such as that they found the approach very novel, interesting and simple (in a good way) and the paper well written and organized.  The reviewers however primarily had significant concerns regarding the empirical evaluation and results.  The authors did not provide a response.

**Reviewer Concerns:**

The authors did not provide a rebuttal.

**Reviewer Scores:**

N/A since there is no rebuttal.

---

### Decision · Program_Chairs · 2026-01-26

Reject